# Integrated Transcriptomics and Metabolomics Provide Insight into Degeneration-Related Molecular Mechanisms of *Morchella importuna* During Repeated Subculturing

**DOI:** 10.3390/jof11060420

**Published:** 2025-05-30

**Authors:** Wenyan Huo, Xuelian He, Yu Liu, Liguang Zhang, Lu Dai, Peng Qi, Ting Qiao, Suying Hu, Pengpeng Lu, Junzhi Li

**Affiliations:** 1Fungal Research Center, Shaanxi Provincial Institute of Microbiology, Xi’an 710043, China; huowenyan0616@126.com (W.H.); xuelhe@163.com (X.H.); ly137261323@126.com (Y.L.); liguang_zhang@163.com (L.Z.); dailutk@126.com (L.D.); qipeng_325@163.com (P.Q.); syjaqt@163.com (T.Q.); husuying0315@163.com (S.H.); luppeng@163.com (P.L.); 2College of Life Science, Shaanxi Normal University, Xi’an 710062, China

**Keywords:** *Morchella importuna*, strain degeneration, non-reducing polyketide synthase, mechanisms

## Abstract

This study investigated *Morchella importuna* strain degeneration during repeated subculturing and employed metabolomics, transcriptomics, and other techniques to explore its molecular mechanisms. Significant metabolic and transcriptional differences were observed between normal mycelia (NM) and degenerated mycelia (DG). Metabolomic analysis revealed 699 differentially expressed metabolites (DEMs) that were predominantly enriched in secondary metabolite biosynthesis pathways, particularly flavonoids and indole alkaloids. Total flavonoid content was markedly higher in NM than in DG, with most flavonoid compounds showing reduced levels in degenerated strains. Transcriptomic profiling revealed 2691 differentially expressed genes (DEGs), primarily associated with metabolic pathways and genetic information processing. Integrated analysis showed that metabolic dynamics were regulated by DEGs, with pyruvate metabolism being significantly enriched. The FunBGCeX tool identified biosynthetic gene clusters (BGCs) in the *M. importuna* genome, highlighting the critical role of the non-reducing polyketide synthase (NR-PKS) gene in flavonoid biosynthesis. This gene exhibited significantly downregulated expression in DG strains. These findings indicate that *M. importuna* degeneration resulted from systemic dysregulation of gene expression networks and metabolic pathway reorganization. The results presented herein also provide theoretical insights into degeneration mechanisms and potential prevention strategies for this edible fungus.

## 1. Introduction

Repeated subculturing is widely used to maintain strains in the production of edible fungi. In order to maintain the vitality of the mycelium, subcultures are usually carried out every 3–4 months [1]. However, this common practice often results in strain degeneration. Degeneration manifests itself in various ways, such as slowed mycelial growth, thin and sparse mycelium, uneven colony margins, altered pigmentation, reduced yield, deteriorated quality, weakened vitality, and reduced stress resistance [1,2]. Studies on *Volvariella volvacea* show that with repeated subculturing, mycelial growth rate and biomass initially increase and then decrease with the number of subculture generations [1]. By the 6th generation, there is a significant decrease in both mycelial growth rate and biomass compared to the original strain. Similar research on *Cordyceps militaris* shows that the yield of *C. militaris* begins to decline by the 4th generation, and by the 6th generation, fruiting bodies can no longer be formed [2]. These results suggest that the widely used method of repeated subculturing in edible mushroom production can lead to strain degeneration. This poses a significant challenge to strain conservation, causes considerable economic losses and hinders the development of the edible fungi industry.

The causes of strain degeneration in edible fungi are diverse, and the specific mechanisms remain unclear. However, existing studies have indicated that the mechanisms underlying strain degeneration during repeated subculturing of edible fungi primarily involve viral infections [3], decreased capacity for synthesis of intracellular enzymes [4,5,6], accumulation of harmful intracellular substances [7,8,9], and alterations in intracellular genetic and/or epigenetic materials [10,11,12]. Qiu et al. isolated a spherical virus with a diameter of 23 nm as well as four double-stranded RNA fragments from a strain of *Pleurotus ostreatus* [3]. The degenerative traits of the strain were alleviated following elimination of these viruses [3], suggesting that viral infection contributes to the degeneration of edible fungal strains [3]. Analysis of the expression levels of β-glucosidase and laccase genes in degenerated *V. volvacea* strains during repeated subculturing by Zhang et al. using quantitative real-time polymerase chain reaction (qRT-PCR) revealed a continuous decline in the expression levels of these two enzymes as the subculturing time increased [6]. These findings suggest that the degeneration of edible fungal strains may be attributed to the reduced capacity for enzymatic production.

The glutathione peroxidase gene (Gpx) from *Aspergillus nidulans* was introduced into two strains of *C. militaris*: one (Cm01) that produced normal fruiting bodies and another degenerated strain (Cm04) that failed to produce fruiting bodies [7]. When compared with the parental strains, the engineered strains overexpressing the Gpx gene were found to exhibit higher glutathione peroxidase activity and enhanced capacity for intracellular ROS scavenging [7]. In addition, strain Cm04 was observed to regain its ability to produce fruiting bodies [7]. Moreover, DNA methylation was found to alter the genotype of *C. militaris*, thereby inducing its degeneration [10].

*Morchella importuna* is a soil-saprotrophic ascomycete that is also a highly esteemed edible fungus [12,13]. Similarly to other edible fungi, its mycelial strains are prone to degeneration after repeated subculturing [12]; however, the specific mechanisms underlying this degeneration have not been thoroughly investigated. Transcriptomics and metabolomics analyses have suggested that the degeneration caused by repeated subculturing may be associated with cellular senescence, loss of cell membrane integrity, decreased DNA repair capacity, autolysis of cells, and degradation of nucleotides [12].

In this study, comprehensive analyses of metabolomics, transcriptomics and the content of 35 flavonoid compounds in the mycelia of *M. importuna* of the NM and DG groups were performed. Significant differences in gene and metabolite expression patterns were found between the degenerated and normal strains. KEGG enrichment analysis revealed the enrichment of DEMs involved in the biosynthesis of other secondary metabolites, especially flavonoid and indole alkaloid biosynthesis. DEGs were mainly enriched in metabolic pathways, especially those associated with amino acid biosynthesis and genetic information processing, especially ribosome, protein processing in the endoplasmic reticulum, RNA transport and ribosome biogenesis in eukaryotes. Measurement of various flavonoid compounds showed a significant decrease in most flavonoid compounds in the degenerate strain. In addition, an NR-PKS gene potentially involved in flavonoid biosynthesis was identified and its expression level was significantly reduced in the degenerate strain.

## 2. Materials and Methods

### 2.1. Sample Preparation

The initial strain of *M. importuna* M483 (deposited at the Shaanxi Institute of Microbiology) was designated as the NM group. The degenerated strain (DG group) was obtained through repeated subculturing on Potato Dextrose Agar (PDA) medium (comprising 200 g of potato, 20 g of glucose, 20 g of agar, and 1000 mL of distilled water). During subculturing, sterile punches were used to obtain inoculum plugs (5 mm in diameter) from the tips of mycelial growth, which were then inoculated onto the PDA medium. Culturing was conducted at a constant temperature of 23 °C. This process was repeated when the mycelium completely covered the PDA plate (15 cm in diameter). Following 23 subcultures (140 days), a significant reduction in mycelial growth rate and marked changes in colony morphology were observed; therefore, the strain was deemed to have degenerated. During the repeated subcultures, the original strain was stored at −80 °C.

### 2.2. Metabolite Extraction

Two groups (NM, DG) and a total of 12 metabolomic samples were prepared in this study (6 replicates in each group). The method of sample extraction was taken from the literature with slight modifications [14,15]. The specific procedure is as follows: *Morchella* mycelium was inoculated into 50 mL PDB liquid medium (formula: potato 200 g, glucose 20 g, distilled water fixed to 1000 mL) and incubated at 23 °C, 180 rpm with shaking for 5 days. The culture solution was collected and centrifuged at 12,000 rpm for 5 min to obtain the supernatant. Then, 100 μL of the supernatant was mixed with 500 μL of extraction solvent containing internal standard solution (20 mg/L, methanol: acetonitrile, *v*/*v* = 1:1), and sonicated for 10 min in an ice-water bath after vortexing for 30 s. The sample was allowed to stand at −20 °C for 1 h, then centrifuged for 15 min at 12,000 rpm at 4 °C. The treated samples were allowed to stand at −20 °C for 1 h, then centrifuged at 4 °C and 12,000 rpm for 15 min, and 500 μL of supernatant was transferred to an EP tube for vacuum drying. The dried product was reconstituted with 160 μL of reconstituted solvent (acetonitrile: water, *v*/*v* = 1:1), and then sonicated again for 30 s with vortex shaking and 10 min in an ice-water bath, and then the supernatant was collected under the same centrifugation conditions, and 1 μL of the supernatant was extracted for UHPLC-MS/MS analysis.

### 2.3. UHPLC-MS/MS Analysis

Metabolic fingerprinting was performed using an optimized ultra-high performance liquid chromatography-tandem mass spectrometry (UHPLC-MS/MS) protocol adapted from established methods [16]. Analytical workflows were performed at Biomarker Technologies Co., Ltd. (Beijing, China) using a Waters Acquity I-Class PLUS UPLC system connected to a quadrupole time-of-flight (QTOF) mass analyser (Xevo G2-XS, Waters, Milford, MA, USA). Chromatographic separation was performed on an HSS T3 reversed-phase column (2.1 mm × 100 mm, 1.8 μm particle size) using gradient elution (0.4 mL/min flow rate) with a 15 min separation window. The binary mobile phase consisted of (A) 0.1% aqueous formic acid and (B) 0.1% formic acid in acetonitrile, programmed as follows: 2% B (0–0.25 min), linear ramp to 98% B (0.25–10 min), isocratic hold (10–13 min) followed by re-equilibration (13.1–15 min). High-resolution mass detection was performed in dual polarity mode with electrospray ionization parameters calibrated as capillary potential ±2500 V (positive/negative modes), cone voltage 30 V, source thermal control (100 °C ionization zone, 500 °C desolvation interface). Gas dynamics were controlled at 800 L/h desolvation flux and 50 L/h cone gas flow. Full scan MS data were acquired over m/z 50–1200 to ensure comprehensive metabolite coverage.

### 2.4. Metabolite Identification and Quantification

Raw data acquired via a MassLynx V4.2 system (Waters, Milford, MA, USA) were subjected to comprehensive processing using the Progenesis QI 2.0 software (Waters, Milford, MA, USA), which included peak extraction, peak alignment, and additional data processing procedures to obtain accurate qualitative and relative quantitative results [16]. Compound identification was facilitated by referencing the integrated online METLIN database within Progenesis QI and a self-built library from Biomarker Technologies Co., Ltd. (Beijing, China). Theoretical fragment ion identification was performed with parent ion mass deviations maintained within 100 ppm and fragment ion mass deviations maintained within 50 ppm [16].

These metabolites were annotated using the KEGG database [17], HMDB database [18], and LIPIDMaps database [19]. Principal components analysis (PCA) was performed using the FactoMineR package (https://cran.r-project.org/web/packages/FactoMineR/index.html) (accessed on 20 December 2024) [20]. Metabolites with a variable importance in projection (VIP) ≥ 1, a *p*-value < 0.05, and a |Log2(foldchange)| ≥ 1 were considered to be significantly differentially expressed (DEMs). The dplyr R package (https://cran.r-project.org/web/packages/dplyr/index.html) (accessed on 5 June 2024) was employed to conduct a hypergeometric test for enrichment analysis of the KEGG annotation results of DEMs. Pathways with a *p*-value < 0.05 were considered statistically significant.

### 2.5. RNA Extraction and Sequencing

A total of 8 transcriptomic samples were collected, with 4 biological replicates per group. Total RNA was extracted from the samples using TRIzol reagent (Invitrogen, Carlsbad, CA, USA). The quality of extracted RNA was assessed using agarose gel electrophoresis and an Agilent 2100 Bioanalyzer (Agilent, Santa Clara, CA, USA), and RNA integrity was assessed using the RNA Integrity Number (RIN). For ONT RNA-Seq analysis, using the PCR cDNA sequencing kit (SQK-PCS109, Oxford Nanopore Technologies, Oxford, Oxfordshire, UK) and the PCR barcoding kit (SQK-PBK004, Oxford Nanopore Technologies, Oxford, Oxfordshire, UK) to generate 8 cDNA libraries (each with a concentration exceeding 20 ng/μL). These libraries were then sequenced on the Nanopore PromethION platform (Oxford Nanopore Technologies, Oxford, UK) at BioMarker Technology Co. (Beijing, China).

### 2.6. Transcriptomic Analysis

Raw reads with an average quality score lower than 6 and a length shorter than 350 base pairs were eliminated. After that, ribosomal RNA (rRNA) sequences were identified and removed by aligning the reads against an rRNA reference database. Full-length, non-chimeric (FLNC) transcripts were then identified by detecting primer sequences at both ends of the reads.

Next, the clustering of detected FLNCs was detected after mapping to the *M. importuna* reference genome (ASM344463v2) using minimap2 [21]. A consensus isoform was then obtained and polished using pinfish2 [22]. Similarly, consensus FLNCs were mapped to the *M. importuna* reference genome using minimap2. To remove redundant FLNCs, all of the mapped transcripts were collapsed using the cDNA Cupcake package (https://github.com/Magdoll/cDNA_Cupcake/wiki) (accessed on 5 June 2024) with a minimum coverage of 85% and a minimum identity of 90%. Consensus FLNCs with sequence differences at 5′ ends were not considered to be redundant isoforms. To identify novel genes and transcripts, we employed gffcompare (https://ccb.jhu.edu/software/stringtie/gffcompare.shtml) (accessed on 5 June 2024) to compare the FLNC transcripts against known reference transcripts in the genome of *M. importuna* (ASM344463v2). The sequences of all of the transcripts were subsequently aligned against a suite of databases, including the Non-Redundant (NR) Protein Database [23] and Swiss-Prot [24], Gene Ontology (GO) [25], Clusters of Orthologous Groups (COG) [26], Eukaryotic Orthologous Groups (KOG) [27], Pfam [28], Kyoto Encyclopedia of Genes and Genomes (KEGG) [29], and evolutionary genealogy of genes: non-supervised orthologous groups (eggNOG) databases [30]. This comprehensive annotation approach enabled us to obtain detailed functional annotations for all of the transcripts.

Full-length reads were aligned to the reference transcriptome sequence. Reads exhibiting a match quality score > 5 were subsequently utilized for quantification. Expression levels were estimated based on reads per gene/transcript per 10,000 reads mapped. Differential expression analysis between the NM and DG groups was conducted using the DESeq2 R package (version 1.6.3) [31]. To control the false discovery rate, the resulting *p*-values were adjusted using the Benjamani–Hochberg method. Genes identified by DESeq2 with an adjusted false discovery rate (FDR) < 0.01 and a fold change (FC) ≥ 2 were considered differentially expressed. Employing the dplyr R package (https://cran.r-project.org/web/packages/dplyr/index.html) (accessed on 5 June 2024) a hypergeometric test was conducted to perform enrichment analysis on the KEGG annotation results of differentially expressed genes. Pathways with a *p*-value < 0.05 were considered statistically significant. Concurrently, the COG [26] and eggNOG [30] databases were employed to conduct orthologous classification of differentially expressed genes. Visualization of data was performed with R.

### 2.7. Correlation Analysis Between the Metabolome and Transcriptome Data

Metabolomics and transcriptomics were integrated using Pearson correlation coefficients (PCCs). All data were log-transformed prior to analysis, and the correlation between metabolomics and transcriptomics was assessed using the core function in the stats R package (version 4.1.0), with a PCC threshold of 0.95. A nine-quadrant plot was generated using the plyr and ggplot2 R packages [32].

### 2.8. Extraction of Biosynthetic Gene Clusters (BGCs) from Genome Sequence Data of M. importuna

The Fungal Biosynthetic Gene Cluster eXtractor (FunBGCeX) tool was employed to identify all of the biosynthetic gene clusters (BGCs) in the genome of *M. importuna* [33]. The annotated genome sequence of *M. importuna*, provided in GenBank format, was systematically processed to extract BGCs using this computational approach.

### 2.9. Quantitative Real-Time PCR (qRT-PCR) Validation

RNA isolation and RT-PCR were conducted as previously described [34]. Briefly, total RNA extraction was performed using an RNAiso Plus Kit (Takara, Tokyo, Japan), after which reverse transcription was conducted with a HiScript II One-Step RT-PCR Kit (Vazyme, Nanjing, China). Next, quantitative real-time PCR (qRT-PCR) was performed using the CFX Connect Real-Time PCR System (Bio-Rad, Hercules, CA, USA). The 20 µL reaction mixture consisted of 1.0 µL cDNA (15 ng), 0.5 µL primers (10 µM), 10 µL SYBR qPCR Master Mix (Vazyme, Nanjing, China), and 8.0 µL ddH_2_O. The assay included four technical replicates. The CYC3 gene, which was previously validated as a stable reference gene in morels [35], was used as the internal control. The primers used for qRT-PCR are listed in Appendix A. Relative gene expression levels were calculated using the 2^−∆∆Ct^ method [36].

### 2.10. Quantitative Analysis of Flavonoids

The total flavonoid content in the mycelial fermentation broth of *M. importuna* was quantified using a NaNO_2_-Al(NO_3_)_3_-NaOH colorimetric assay. Briefly, the supernatant of fermentation broth was concentrated via freeze-drying to enhance the analyte concentration. Subsequently, a 10 mL aliquot of the freeze-dried concentrate was transferred to a 50 mL volumetric flask. To this, 16 mL of 30% (*v*/*v*) ethanol was added, followed by 2 mL of 5% (*w*/*v*) sodium nitrite. The mixture was then allowed to stand for 5 min, after which 2 mL of 10% (*w*/*v*) aluminum nitrate was added and the solution was allowed to stand for an additional 6 min. The reaction was then completed by adjusting the volume to 50 mL with 4% (*w*/*v*) sodium hydroxide solution and allowing the mixture to stand for a further 10 min. Finally, the absorbance was measured at 510 nm using a UV-Vis spectrophotometer, with a reagent blank serving as the reference. The total flavonoid content was then determined based on a calibration curve constructed using rutin as the reference standard.

The content of different flavonoid compounds in the mycelium of *M. importuna* was determined by means of liquid chromatography-mass spectrometry (LC-MS) [37,38,39]. Briefly, standard solutions were prepared by accurately weighing flavonoid standards and then dissolving them in 80% methanol to create individual stock solutions, which were subsequently mixed in appropriate ratios to obtain working standard solutions. All of the stock and working solutions were stored at −20 °C (see Appendix A for specific conditions). For sample preparation, an appropriate amount of the sample was weighed into a 2 mL centrifuge tube, after which 600 μL of methanol was added and the samples were vortexed for 60 s. Two steel balls were added to the tube, which was then placed in a grinder adapter, immersed in liquid nitrogen for 5 min, and subjected to a freeze–thaw cycle at room temperature. The sample was subsequently shaken at 60 Hz for 1 min. This freeze–thaw and shaking process was repeated at least twice, after which the sample was sonicated for 15 min at room temperature and centrifuged at 12,000 rpm for 5 min at 4 °C. The supernatant was then filtered through a 0.22 μm membrane and transferred to an LC-MS vial for analysis. Next, chromatographic separation was performed using an ACQUITY UPLC^®^ BEH C18 column (2.1 mm × 100 mm, 1.7 μm, Waters, USA) with an injection volume of 5 μL and a column temperature of 40 °C. The mobile phase consisted of 0.1% formic acid in water (A) and methanol (B) applied at a flow rate of 0.25 mL/min, with gradient elution conditions as follows: 0–1 min, 10% B; 1–3 min, 10–33% B; 3–10 min, 33% B; 10–15 min, 33–50% B; 15–20 min, 50–90% B; 20–21 min, 90% B; 21–22 min, 90–10% B; and 22–25 min, 10% B. Mass spectrometry was conducted using an electrospray ionization (ESI) source in negative ionization mode, with an ion source temperature of 500 °C and a voltage of −4500 V. The collision gas pressure was maintained at 6 psi, the curtain gas pressure was maintained at 30 psi, and the pressure of both the nebulizer and auxiliary gas was maintained at 50 psi, with detection performed in multiple reaction monitoring (MRM) mode.

## 3. Results

### 3.1. Obtaining of the Degenerate Strain and Metabolomics Analysis of Normal and Degenerate Mycelia of M. importuna

After 140 days of repeated subculturing, the resulting strain exhibited notable degeneration characteristics, such as a marked decline in mycelial growth rate, irregular colony margins, reduced sclerotium production, and a lighter color. Especially during fermentation on PD medium, compared with the NM group, the mycelium and extracellular metabolites of the DG group were both lighter in color, particularly the extracellular metabolites (Figure 1a,b). To investigate the differences between normal and degenerated mycelium of *M. importuna* (Figure 1a,b), non-targeted metabolomics (UHPLC-MS/MS) analysis was used to analyze the metabolomes of these two groups of samples. Correlation analysis of the QC samples showed that the metabolomics data in this study were highly stable and reliable Appendix A.

A total of 1884 metabolites were annotated from the metabolome samples of morel mycelia (Appendix A). Among these, 770 compounds were classified into eight major categories: lipids and lipid-like molecules, organic acids and derivatives, organoheterocyclic compounds, organic oxygen compounds, benzenoids, nucleosides, nucleotides and analogs, phenylpropanoids and polyketides, and organic nitrogen compounds. Lipids accounted for the highest proportion with 238 compounds, while organic nitrogen compounds were the least abundant with only 12 compounds (Figure 1d). PCA (Figure 1c) and Pearson correlation analysis Appendix A showed that mycelial metabolites from different groups formed distinct clusters, indicating that normal and degenerated mycelium have different characteristic metabolites.

Metabolites with VIP scores ≥ 1, |Log2(fold change)| ≥ 1, and *p*-values < 0.05 were identified as DEMs through differential expression analysis and PLS-DA analysis. A total of 699 DEMs were screened, including 296 up-regulated and 403 down-regulated metabolites (Figure 2a,b and Appendix A). The top 20 up- and down-regulated DEMs between NM and DG groups are shown in Appendix A.

Metabolic pathways and functions of DEMs were enriched using KEGG (Appendix A). The results showed that these DEMs were mainly enriched in pathways related to the biosynthesis of secondary metabolites, particularly the biosynthesis of flavonoids and indole alkaloids (Appendix A and Figure 2c). These results indicated that the secondary metabolites, especially flavonoids and indole alkaloids, in the degenerated mycelium of *M. importuna* have undergone significant changes when compared with the NM group.

### 3.2. Transcriptomics Analysis Between Normal and Degenerate Mycelia of M. importuna

To investigate the molecular mechanisms underlying metabolic differences between normal and degenerated mycelia of *M. importuna*, comparative transcriptome analysis was performed using nanopore sequencing on eight biological samples (four per group). Sequencing generated 47,666,572 high-quality reads with average read lengths ranging from 1036 to 1256 base pairs Appendix A. Subsequent processing identified 26,804,721 full-length non-concatemer (FLNC) reads containing intact primer sequences at both termini, representing 82.94–85.71% of total reads across samples Appendix A. High mapping efficiency was observed, with 87.08–94.48% of FLNCs successfully aligned to reference sequences Appendix A. Post-clustering refinement and error correction yielded 246,896 consensus transcripts (24,039–37,512 per sample; Appendix A). Following deduplication and quality filtering, a final set of 23,932 unique FLNC transcripts was established for downstream analysis.

Transcript and gene expression quantification was performed using counts per million (CPM) normalization. Inter-sample reproducibility was assessed through Pearson correlation analysis, revealing strong consistency across biological replicates (Figure 3b), with all pairwise comparisons demonstrating high concordance (coefficient of determination (R^2^ > 0.75). Principal component analysis (PCA) further validated sample clustering patterns, showing tight grouping of replicates within the normal mycelia (NM) and degenerated mycelia (DG) groups, alongside clear spatial separation between the two experimental conditions, indicative of systematic transcriptomic divergence (Figure 3a). To assess the reproducibility of expression, the CPM distribution was visualized using box plots, which showed that the expression levels within the NM and DG sample groups were comparable. Notably, the NM group exhibited significantly elevated global expression levels relative to the DG group (Figure 3c). These analytical outcomes collectively confirm the reliability of the transcriptomic datasets, including adequate sequencing depth and technical reproducibility, thereby supporting robust downstream functional investigations.

Differential expression analysis employing stringent thresholds (|Log2(foldchange)| ≥ 1, FDR < 0.01) revealed 2691 DEGs between NM and DG groups, comprising 1592 upregulated and 1099 downregulated candidates (Figure 3d and Appendix A). Functional annotation through KEGG pathway analysis indicates that these genes are mainly involved in core metabolic regulation, particularly amino acid biosynthetic pathways, and also play key roles in genetic information processing pathways, including ribosome, protein processing in endoplasmic reticulum, RNA transport and ribosome biogenesis in eukaryotes (Figure 3e, and Appendix A). eggNOG orthologous classification analysis of DEGs revealed that the following function classes contained the most DEGs (in order): carbohydrate transport and metabolism, secondary metabolites biosynthesis, transport and catabolism, and post-translational modification, protein turnover, chaperones Appendix A. These classification results are consistent with the results of KEGG enrichment analysis.

### 3.3. Integrated Metabolomics and Transcriptomics Analysis

Integrative analysis of transcriptional and metabolic networks was performed by stringent multi-omics correlation profiling (Pearson correlation coefficient thresholded at |r| > 0.95). The non-agonal coordinate system visualized coordinated fluctuations between 1592 upregulated/1099 downregulated genes (|log2FC| ≥ 1, FDR < 0.01) and their associated metabolites (|log2FC| ≥ 1). In this Cartesian framework, co-regulated molecular pairs clustered in diagonal sectors (positive correlation: quadrants 3 and 7; inverse relationships: quadrants 1 and 9), while baseline expression units populated the central zone (Figure 4a). This spatial organization revealed a tight interconnectivity between transcriptional reprogramming and metabolic restructuring, suggesting candidate regulatory nodes for phenotype modulation. Furthermore, multi-omics pathway convergence analysis using KEGG annotation identified the pyruvate node of central carbon metabolism as a key integration hub, demonstrating coordinated enrichment patterns for both differential transcripts and metabolites (Figure 4b).

### 3.4. Extraction of BGCs and qRT-PCR Analysis of NR-PKS Gene

The FunBGCeX tool was used to identify novel fungal BGCs from the genome of *M. importuna*. A total of 17 BGCs were mined, among which BGC4 contains a non-reducing polyketide synthase (NR-PKS) composed of 2152 amino acids Appendix A. This NR-PKS shares 36.3% sequence identity with the oosporein synthase 1 (OpS1) from *Beauveria bassiana* and encompasses the following specific domains: SAT-KS-AT-PT-ACP-ACP-TE (SAT: Starter unit Acyl Transferase; KS: Keto Synthase; AT: Acyl Transferase; PT: Product Template; ACP: Acyl Carrier Protein; TE: Thioesterase) Appendix A [40]. Quantitative real-time PCR (qRT-PCR) was employed to assess the expression levels of this NR-PKS gene (gene ID: gene-LAJ45_06308) in normal and degenerated strains. The results demonstrated that the expression level of the NR-PKS gene in the degenerated strain showed a significant decrease (*p* < 0.001, Figure 5a) relative to the NM group, corroborating the findings of full-length transcriptome sequencing analysis (*p* < 0.001, Figure 5b).

### 3.5. Determination of Flavonoid Content

To assess the impact of strain degeneration on flavonoid production, the total flavonoid content in the mycelial fermentation broth of the NM and DG groups was measured using the NaNO_2_-Al (NO_3_)_3_-NaOH colorimetric assay. The contents of 35 flavonoid compounds were also determined by LC-MS. The results showed that the total flavonoid content in the mycelial fermentation broth of the NM group was significantly higher (approximately 1.43 times) than that of the DG group Appendix A. Of the 35 flavonoid compounds, 17 were detected in the NM group and 14 in the DG group Appendix A. The NM group had significantly higher levels of 15 compounds than the DG group (*p* < 0.05, Table 1). NM group levels of Chrysin, Liquiritigenin, Naringenin, Quercetin, Dihydromyricetin, Vitexin, Quercitrin, Quercetin 3—glucoside, Naringin, and Rutin were over three times higher than those in the DG group (Table 1).

## 4. Discussion

Repeated subculturing can cause the degeneration of edible fungus strains, which is manifested by a decline in mycelial growth rate, alterations in pigment production (either increase or decrease), and irregular colony margins [1,2]. Existing research has shown that repeated subculturing can also induce degeneration of the strains of *M. importuna*, but the characteristics of degeneration in the mycelial growth stage are not well defined. Our experimental results demonstrated that the strains of *M. importuna* exhibited characteristics such as a decline in mycelial growth rate, reduced sclerotium production, and decreased pigment production during repeated subculturing.

With the rapid development of high-throughput sequencing technology, it has been applied to investigations of the degeneration mechanisms of edible fungi during subculturing. The strain *Cordyceps militaris* was continuously subcultured for six generations by Yin et al. [2], and the transcriptomes of strains of each generation were sequenced. The results showed that the degeneration of *C. militaris* was mainly related to toxin biosynthesis, energy metabolism, DNA methylation, and chromatin remodeling. In our study, third-generation transcriptome sequencing (Nanopore) and analysis were performed on normal and degenerated mycelia of *M. importuna*. KEGG enrichment analysis revealed that the DEGs were mainly enriched in metabolic pathways (particularly amino acid biosynthesis) and genetic information processing (especially ribosome, protein processing in the endoplasmic reticulum, RNA transport, and ribosome biogenesis in eukaryotes) (Figure 3e and Appendix A). The eggNOG orthologous classification analysis of DEGs also showed that the following functional classes contained the most DEGs in the following order: carbohydrate transport and metabolism, secondary metabolites biosynthesis, transport and catabolism, and posttranslational modification, protein turnover, and chaperones Appendix A. Thus, it was speculated that translation, posttranslational modification, and metabolic network reconstruction, particularly of the secondary metabolic network, were closely associated with degeneration of *M. importuna*.

Metabolomics can also be applied to investigation of the degeneration mechanisms of edible fungal strains. Chen et al. studied the effects of different preservation methods on the major metabolites, and potential metabolic pathways of *M. importuna* were studied based on non-targeted metabolomics [12]. The results revealed that the metabolic pathways of *M. importuna* were greatly influenced by different preservation methods. For instance, the metabolites were found to contain higher levels of antioxidants such as kojic acid and thymol in treatment T2 (preserved on a soil and sawdust mixed medium (LSS) without subculturing), which suggests that nutrient-limited preservation conditions can help reduce oxidative stress [12]. In addition, the metabolites contained higher levels of xanthine in the CK treatment (preserved on PDA medium without subculturing), which indicates that repeated subculturing may result in cell autolysis and nucleotide degradation, thus leading to degeneration [12].

In this study, the DEMs of normal and degenerated mycelia of *M. importuna* were subjected to KEGG enrichment analysis based on non-targeted metabolomics. The results showed that these DEMs were mainly enriched in the biosynthesis of other than the main metabolites (Biosynthesis of other secondary metabolites), particularly flavonoid and indole alkaloid biosynthesis (Figure 2c and Appendix A). To monitor changes in the levels of different flavonoid compounds during strain degeneration, the contents of 35 flavonoid compounds in the mycelia of NM and DG groups of *M. importuna* were measured using LC-MS. The results revealed that the levels of 15 flavonoid compounds were significantly decreased in the degenerated strain. Furthermore, integrated metabolomics and transcriptomics analysis was performed to identify key genes involved in the biosynthesis of flavonoid compounds in *M. importuna*. The findings indicated that metabolic dynamics might be subject to regulation, either directly or indirectly, by the corresponding DEGs. Both DEMs and DEGs were significantly enriched in pyruvate metabolism. However, based on the current database, no genes or gene clusters related to flavonoid biosynthesis were annotated. To address this, the FunBGCeX tool [33] was utilized to identify novel fungal BGCs within the genome of *M. importuna*. Among the 17 BGCs identified, an NR-PKS composed of 2152 amino acids was detected Appendix A. This NR-PKS shares 36.3% sequence identity with OpS1 from *Beauveria bassiana* [40]. Both OpS1 and NR-PKS from *M. importuna* contain domains such as KS, AT, ACP, and TE [40]. Zhang et al. [41] discovered a nonribosomal peptide synthetase-polyketide synthase (NRPS-PKS) in *Aspergillus candidus*. This enzyme is responsible for the generation of the key precursor chalcone. Then, a novel chalcone isomerase (CHI) catalyzes the conversion of chalcone into flavanone. Finally, a novel flavone synthase (FNS), which is flavin mononucleotide (FMN)-dependent oxidoreductase, catalyzes the desaturation of flavanone to flavone. It is interesting that NRPS-PKS in *A. candidus* and NR-PKS from *M. importuna* both also contain the domains KS, AT, ACP, and TE [41]. Thus, it is hypothesized that the NR-PKS gene is a key gene involved in the catalysis of flavonoids by *M. importuna*. qRT-PCR and RNA-seq results revealed that the level of NR-PKS gene expressed in the degenerated *M. importuna* strain was significantly lower than that in the normal strain. Metabolomic analysis also showed a marked decrease in flavonoid compound content in the degenerated strain. These results further confirm that the NR-PKS gene of *M. importuna* is a crucial gene for the synthesis of various flavonoid compounds and suggest a close relationship between flavonoid compound content and strain degeneration of *M. importuna*.

Flavonoids are natural antioxidants that can scavenge intracellular free radicals (e.g., ROS), reduce oxidative damage, and protect cells from oxidative stress [42]. Previous studies have shown that the degeneration of *C. militaris* is linked to cellular oxidative stress and that transformation of an antioxidant glutathione peroxidase gene could increase fungal resistance against oxidative stress and thus reverse fungal degeneration in cultures [7]. Other studies have shown that nutrient-limited storage conditions can alleviate cellular oxidative stress, maintain antioxidant capacity, and preserve the high productivity of *M. importuna* [12]. Thus, it is inferred that intracellular flavonoids in *M. importuna* can reduce oxidative damage by eliminating ROS. A decline in their levels may weaken mycelial antioxidant capacity, cause oxidative damage, and trigger strain degeneration.

## 5. Conclusions

In summary, it can be inferred that degeneration of the strains of *M. importuna* during repeated subculturing was caused by multiple factors, including (1) systemic dysregulation of gene expression networks due to reconfiguration of translation and post-translational modification and (2) changes in metabolic products from metabolic network reconfiguration, especially secondary metabolites like flavonoids. Notably, the decrease in intracellular flavonoids from metabolic network reconfiguration weakens mycelial antioxidant capacity, causes oxidative damage, and leads to the degeneration of *M. importuna*. Therefore, frequent subculturing should be avoided during the preservation of *M. importuna*, and methods like low-temperature dormancy or adding antioxidants can slow degeneration. However, the specific functions of the NR-PKS gene in preventing degeneration of *M. importuna* and its expression regulation mechanisms require further investigation. As differences in intracellular flavonoid levels may also exist between normal and degenerate strains, further exploration into the alterations of intracellular metabolites, particularly flavonoids, during strain degeneration is warranted. This would enhance our comprehensive understanding of the mechanisms underlying strain degeneration in *M. importuna*.

## Figures and Tables

**Figure 1 jof-11-00420-f001:**
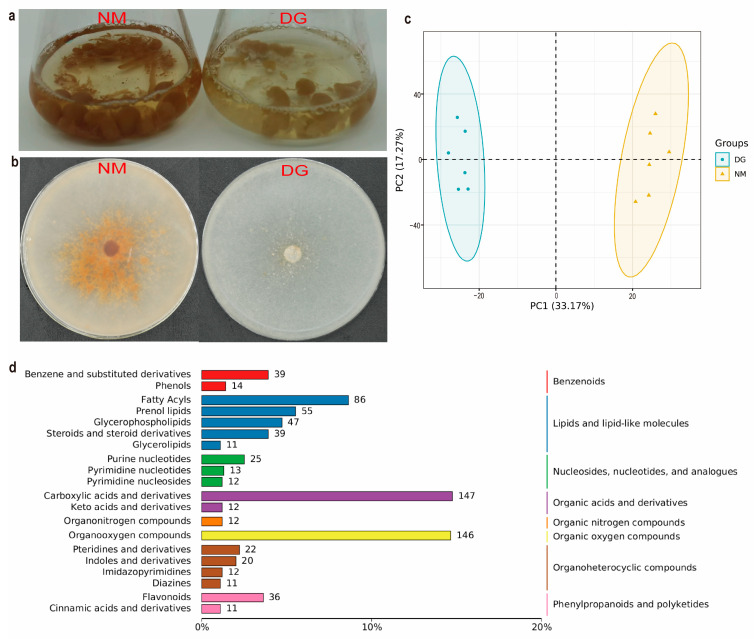
Morphological characteristics and comparative overview of all of the metabolites of *Morchella importuna*. Photos of representative phenotypes of normal (NM) and degenerate (DG) mycelia of *M. importuna* grown on Potato Dextrose (**a**) and Potato Dextrose Agar (**b**) media. (**c**) PCA of metabolite compositions between NM and DG mycelia of *M. importuna*. (**d**) Classification of metabolites.

**Figure 2 jof-11-00420-f002:**
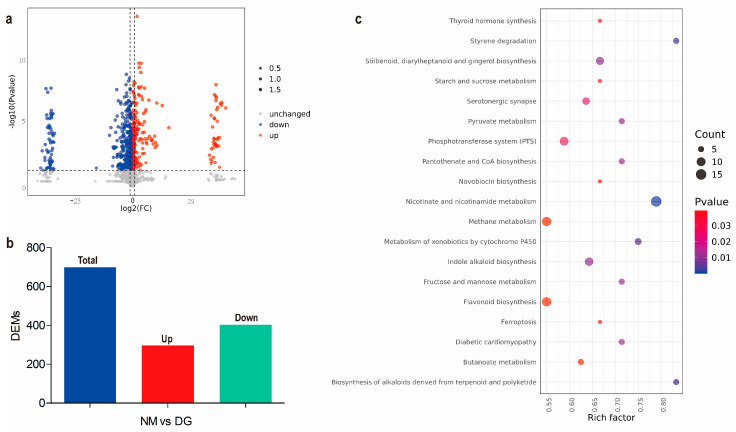
Metabolomics analysis between normal and degenerate mycelia of *Morchella importuna*. (**a**) Volcano plot of the DEMs of NM and DG groups. (**b**) Statistical analysis of differentially expressed metabolites (DEMs). Total: total DEMs; Up: upregulated metabolites; Down: downregulated metabolites. (**c**) KEGG enrichment analysis of the DEMs (Counts > 3) in NM vs. DG. The size of the dots represents the number of metabolites, while the color represents the *p*-value.

**Figure 3 jof-11-00420-f003:**
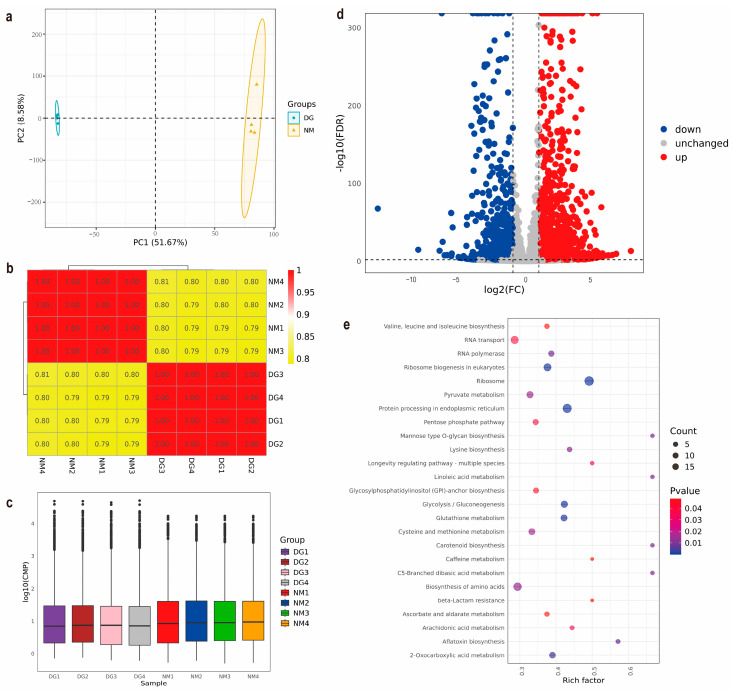
Transcriptomics analysis between normal and degenerate mycelia of *Morchella importuna*. (**a**) PCA of the gene expression profiles of all of the samples. (**b**) Pearson correlation analysis of all of the transcriptome samples. (**c**) Distribution of CPM values across samples. (**d**) Volcano plot of the DEGs of NM and DG groups. (**e**) KEGG enrichment analysis of the DEGs in NM vs. DG. The size of the dots represents the number of genes, while the color represents the *p*-value.

**Figure 4 jof-11-00420-f004:**
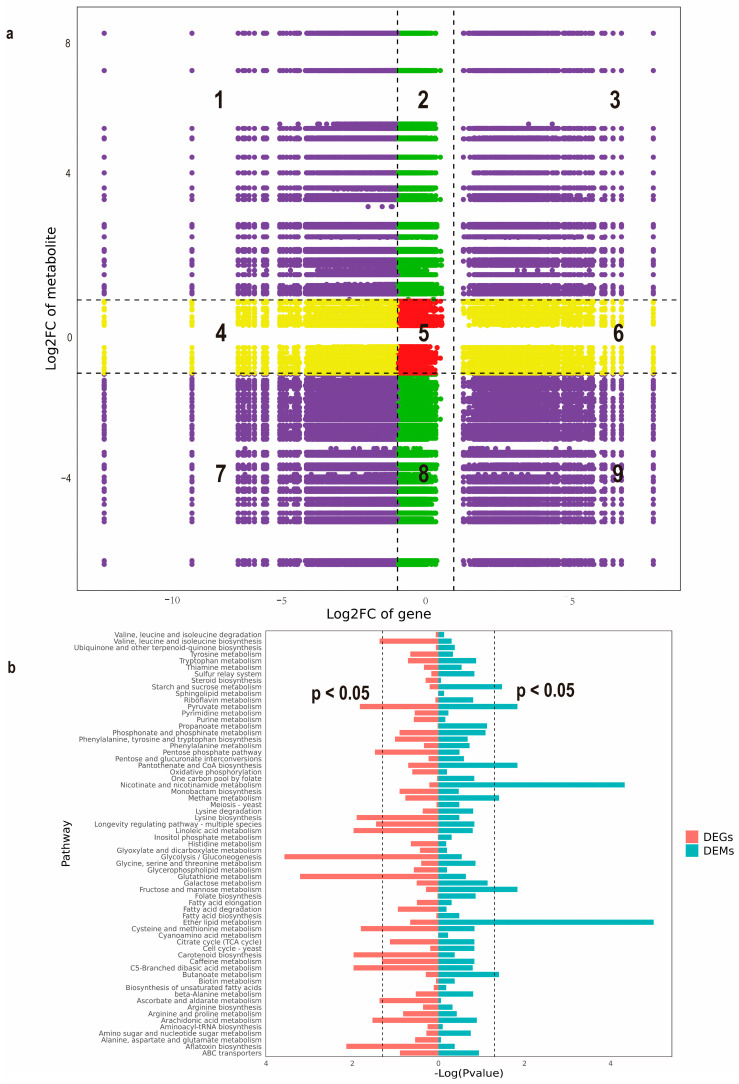
Integrated metabolomics and transcriptomics analysis of *Morchella importuna*. (**a**) Nine quadrant plots for correlation analysis between NM and DG. Numbers 1–9 represent different quadrants, respectively. Purple indicates both DEGs and DEMs show significant changes (|Log2(foldchange)| ≥ 1); Green indicates only DEMs show significant changes (|Log2(foldchange)| ≥ 1) while DEGs show non-significant changes (|Log2(foldchange)| < 1); Yellow indicates only DEGs show significant changes (|Log2(foldchange)| ≥ 1) while DEMs show non-significant changes (|Log2(foldchange)| < 1); Red indicates neither DEGs nor DEMs show significant changes (|Log2(foldchange)| < 1). (**b**) KEGG pathway enrichment analysis of the DEGs and DEMs between NM and DG.

**Figure 5 jof-11-00420-f005:**
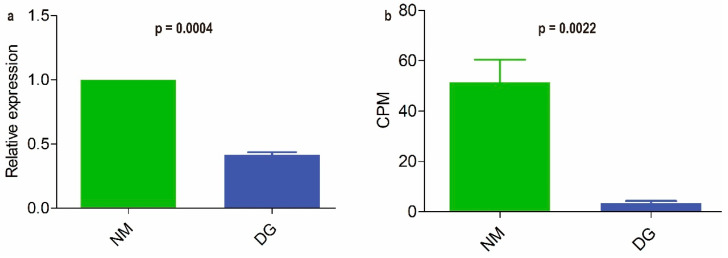
Analysis and qRT-PCR validation of NR-PKS gene from *Morchella importuna*. Relative expression levels of the NR-PKS gene obtained by qRT-PCR (**a**); Counts per million (CPM) values from nanopore sequencing (**b**). *T* tests were used to identify significant differences between the relative expression level and CPM value. Each value is presented as the mean ± SD.

**Table 1 jof-11-00420-t001:** Content of flavonoid compounds of samples from the normal mycelia (NM) and degenerated mycelia.

Compounds	Content (ng/g)		Fold Change	*p* Value
	NM	DG		
Chrysin	402.6 ± 5.604	29.99 ± 1.903	13.424	***
Liquiritigenin	381.7 ± 10.13	51.95 ± 0.998	7.347	***
Apigenin	0.592 ± 0.04	0.71 ± 0.036	0.834	*
Naringenin	54.7 ± 0.685	12.15 ± 0.635	4.502	***
Luteolin	0.673 ± 0.091	0.255 ± 0.016	2.639	*
L-Epicatechin	102.4 ± 3.569	ND	ND	
Kaempferide	1.38 ± 0.052	0.495 ± 0.029	2.788	***
Quercetin	138.4 ± 6.425	36.08 ± 2.34	3.836	***
Dihydromyricetin	150.9 ± 3.777	47.5 ± 1.575	3.177	***
Vitexin	1.326 ± 0.055	0.387 ± 0.018	3.426	***
Astragalin	15.91 ± 0.777	18.78 ± 0.565	0.847	**
Quercitrin	8.374 ± 0.606	0.245 ± 0.011	34.180	***
Cynaroside	1.791 ± 0.077	ND	ND	
Quercetin 3-glucoside	7.514 ± 0.496	1.611 ± 0.107	4.664	***
Naringin	212.2 ± 4.018	41.38 ± 2.199	5.128	***
Diosmin	3.226 ± 0.079	ND	ND	
Rutin	342.5 ± 2.849	76.86 ± 0.248	4.456	***

*T* tests were used to evaluate the significant differences. Each value is presented as the mean ± SD. * *p* < 0.05, ** *p* < 0.01, *** *p* < 0.001. ND: Not Detected.

## Data Availability

The raw ONT sequencing reads have been deposited to the CNCB (China National Center for Bioinformation) GSA database under the BioProject ID of PRJCA038423 (BioSample accession: subSAM139053).

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
