# Peer review of "Integrated Transcriptomics and Metabolomics Provide Insight into Degeneration-Related Molecular Mechanisms of Morchella importuna During Repeated Subculturing"

_jof, 2025, doi:10.3390/jof11060420_

Round 1
Reviewer 1 Report
Degeneration and genetic mutations are common in fungal strains kept in culture for a long time, and therefore research of the present manuscript is needed, There is a thorough research on degeneration of cultivated Morchella importuna by Chen et al. 2021 (Microb Biotechnol 2025 Jan;18(1):e70045.doi: 10.1111/1751-7915.70045) which analyses the effect of long term preservation on the transcriptomics and metabolomics of M. importuna and the effect on fruiting capacity. The authors know this publication well but point out that repeated subculturing is not dealt with in the publication and it can also induce degeneration of the strains of M. importuna. Although the goal of the manuscript is to clarify how the repeated subculture changes the strain, the information about the time and the number of subcultures to obtain the DG strain is missing. In addition, no information is available how the MN strain was kept during subculturing of the DG strain. The description of the phenotype changes of the DG strain is not very detailed. Only Fig. 1a an b.
The metabolic and transcriptional analysis as well as analysis and comparison of flavonoids in MN and DG strains are done with skill as far as I am able to deduce. The results are not very exciting: (1) the pyruvate node as a key integration hub for demonstrating coordinated enrichment patterns for both differential transcripts and metabolites, (2) differences in flavonoid quality and quantity between NM and DG strains as well as (2) considering the NR-PKS gene as key gene regulating flavonoid biosynthesis. Might be that the degeneration of strain was not really profound and the described differences are perhaps mild symptoms of degeneration. I wonder did the authors try the recovery of the DG strain.
No comments
Author Response
Comments 1: Degeneration and genetic mutations are common in fungal strains kept in culture for a long time, and therefore research of the present manuscript is needed, There is a thorough research on degeneration of cultivated Morchella importuna by Chen et al. 2021 (Microb Biotechnol 2025 Jan;18(1):e70045.doi: 10.1111/1751-7915.70045) which analyses the effect of long term preservation on the transcriptomics and metabolomics of M. importuna and the effect on fruiting capacity. The authors know this publication well but point out that repeated subculturing is not dealt with in the publication and it can also induce degeneration of the strains of M. importuna. Although the goal of the manuscript is to clarify how the repeated subculture changes the strain, the information about the time and the number of subcultures to obtain the DG strain is missing. In addition, no information is available how the MN strain was kept during subculturing of the DG strain. The description of the phenotype changes of the DG strain is not very detailed. Only Fig. 1a an b.
Response 1: Thank you for pointing this out. I agree with this comment. Therefore, I have addressed the reviewers' comments in the revised manuscript; specific changes can be found on lines 101 – 104, 276 – 280.
Comments 2: The metabolic and transcriptional analysis as well as analysis and comparison of flavonoids in MN and DG strains are done with skill as far as I am able to deduce. The results are not very exciting: (1) the pyruvate node as a key integration hub for demonstrating coordinated enrichment patterns for both differential transcripts and metabolites, (2) differences in flavonoid quality and quantity between NM and DG strains as well as (3) considering the NR-PKS gene as key gene regulating flavonoid biosynthesis. Might be that the degeneration of strain was not really profound and the described differences are perhaps mild symptoms of degeneration. I wonder did the authors try the recovery of the DG strain.
Response 2: Thank you for pointing this out. We did not attempt to restore the DG strain but confirmed its significant degeneration. To verify the stability of degeneration, we continuously cultured the strain on PDA medium for three generations and found that it consistently exhibited degeneration features, thus confirming its degeneration state. In addition, when we tested the degenerated strain, it was found to be completely sterile.
Reviewer 2 Report
The authors have carried out an integrative transcriptomic and metabolomic analysis of normal and degenerated strains to see what genomic or metabolomic changes are correlated with the phenotypic changes seen in strains after repeated subculturing.|
The manuscript is well written and experiments carried out well too.
There are a number of issues, however, that might need some clarifications.
The manuscript describes only the study of metabolites that are secreted into the medium (supernatant was samples of the centrifuged culture). There are also metabolites that are not secreted. According to Wu et al 2022 for example, flavonoids are produced intra and extracellular by Aspergillus. The have different functions. Why were only extracellular flavonoids studied?
In line 274-275, the authors mention 1884 metabolites annotated from the metabolome of morel mushrooms. Are these data generated from previous experiments, and if so, by who (ref.)? Add this information into figure 1. Since this manuscript describes metabolites excreted by mycelium, how comparable are mushroom data with the mycelia data?
Line 489-490. “..however, no genes………..detected”. Were these genes not present in the genome or were they not represented in the DEG analysis?
Lines 495-498. The authors hypothesize that a NR-PKS gene is a key gene in the synthesis of flavonoids. Zhang et al 2023. (Discovery of a Unique Flavonoid Biosynthesis Mechanism in Fungi by Genome Mining), mention 2 other key genes typical for fungal biosynthesis of flavonoids: chalsome isomerase and flavone synthase). Are these gene present in the Morchella genome? Also, citing this paper would be useful.
The authors suggest that an adapted media for subculturing might solve the issue of strain degeneration. In many other preservation of mushroom cultures, high number of samples are stored under liquid nitrogen. That allows the recovery of normal strains from nitrogen, each time new spawn has to be made. Is this not done for Morchella or is this fungus difficult to store in liquid nitrogen?
Line 479: “ …….. biosynthesis of other secondary metabolites…” Why “other”?
Lines 277-278: here are some misplaced hyphens present.
The text within the blocks of figure 3b is nor readable.
Author Response
Comments 1: The manuscript describes only the study of metabolites that are secreted into the medium (supernatant was samples of the centrifuged culture). There are also metabolites that are not secreted. According to Wu et al 2022 for example, flavonoids are produced intra and extracellular by Aspergillus. The have different functions. Why were only extracellular flavonoids studied?
Response 1: Thank you for pointing this out. During the experiment, I focused on determining the extracellular flavonoid content. This decision was primarily based on the observation that, during fermentation, the mycelium and secretions of the NM and DG strain groups underwent color changes, with the secretions exhibiting a more pronounced color alteration. Consequently, I prioritized the measurement of extracellular flavonoid compounds. Your suggestion has been extremely valuable. In the subsequent research, I plan to incorporate the intracellular flavonoid content into the study as well to gain a more comprehensive understanding of the relevant phenomena.
Comments 2: In line 274-275, the authors mention 1884 metabolites annotated from the metabolome of morel mushrooms. Are these data generated from previous experiments, and if so, by who (ref.)? Add this information into figure 1. Since this manuscript describes metabolites excreted by mycelium, how comparable are mushroom data with the mycelia data?
Response 2: Thank you for pointing this out. These metabolites were annotated using the KEGG database, HMDB database, and LIPIDMaps database. The methodological descriptions are in lines 150 - 151 of the manuscript. The detailed information of the 1884 metabolites is in Table S3.
Comments 3: Line 489-490. “..however, no genes………..detected”. Were these genes not present in the genome or were they not represented in the DEG analysis?
Response 3: Thank you for pointing this out. Based on the current database, no genes or gene clusters related to flavonoid biosynthesis were annotated (probably because the existing database has no such genes and gene clusters). To address this, the FunBGCeX tool was utilized to identify novel fungal BGCs within the genome of M. importuna. For the detailed description of these revisions, please refer to lines 510 - 513 of the manuscript.
Comments 4: Lines 495-498. The authors hypothesize that a NR-PKS gene is a key gene in the synthesis of flavonoids. Zhang et al 2023. (Discovery of a Unique Flavonoid Biosynthesis Mechanism in Fungi by Genome Mining), mention 2 other key genes typical for fungal biosynthesis of flavonoids: chalsome isomerase and flavone synthase). Are these gene present in the Morchella genome? Also, citing this paper would be useful.
Response 4: Thank you for pointing this out. In the genome of M. importuna, the two above genes were not identified. This NR-PKS shares 36.3% sequence identity with OpS1 from Beauveria bassiana, and both OpS1 and NR-PKS from M. importuna contain domains such as KS, AT, ACP, and TE. Given these similarities and domain compositions, the article should appropriately reference the study by Feng et al. on the OpS1 gene (Fungal biosynthesis of the bibenzoquinone oosporein to evade insect immunity).
Comments 5: The authors suggest that an adapted media for subculturing might solve the issue of strain degeneration. In many other preservation of mushroom cultures, high number of samples are stored under liquid nitrogen. That allows the recovery of normal strains from nitrogen, each time new spawn has to be made. Is this not done for Morchella or is this fungus difficult to store in liquid nitrogen?
Response 5: Thank you for pointing this out. In practical production, due to cost considerations, mushroom producers prefer repeated subculturing to liquid nitrogen preservation. Based on the above considerations, I suggested that an adapted media for subculturing might solve the issue of strain degeneration.
Comments 6: Line 479: “ …….. biosynthesis of other secondary metabolites…” Why “other”?
Response 6: Thank you for pointing this out. “Biosynthesis of other secondary metabolites” is the name of a dedicated metabolic pathway in the KEGG database. see the figure below.
Reviewer 3 Report
Please see the attachment.
Please see the attachment.

Author Response
Comments 1: Supplementary files: Externally hosted supplementary files, Supplementary 1, Doi: 10.5281/zenodo.15188185: This DOI cannot be found in the DOI System.
Response 1: Thank you for pointing this out. I have submitted the Supplementary Files in the submission system without the need for a DOI system.
Comments 2: A rather intriguing observation is that DEMs are predominantly enriched in secondary metabolite biosynthesis pathways linked to flavonoids. Concurrently, the reduced level of most flavonoid compounds is shown in degenerated strains. Do you think that it is a repeated subculturing that is an important reason for failure in finding flavonoids in fungi?
Some authors believe that this class of compounds does not exist in mushrooms, i.e. in both basidiomycetes and ascomycetes. For instance, Gil-Ramírez et al. [Gil-Ramírez et al., 2016] claim that "the flavonoids identified in several mushroom species using advanced identification devices such as DAD and MS might be due to sample contaminations". Please extend a discussion of this (apparent?) contradiction.
Response 2: Thank you for pointing this out. Some authors believe that flavonoids does not exist in mushrooms, which is an interesting viewpoint. While this might hold true for mushroom fruiting bodies, it's plausible that during other stages of their life cycle, such as the mycelial stage, mushrooms do produce flavonoids. This is supported by Zhang's discovery of two key genes for fungal flavonoid biosynthesis: chalcone isomerase and flavone synthase (Discovery of a Unique Flavonoid Biosynthesis Mechanism in Fungi by Genome Mining). Thus, I believe that repeated subculturing isn't the main reason for the difficulty in detecting flavonoids in fungi. Mushrooms are capable of producing flavonoids, because they possess the key genes for flavonoids biosynthesis.
Comments 3: Line 277: “Organohet-erocyclic compounds” should be written as “Organoheterocyclic compounds”.Line 278: “nitrogen”.
Response 3: Thank you for pointing this out. I have corrected the spelling mistakes in the manuscript as per your request. The specific revisions can be found on Lines 289 and 290.
Round 2
Reviewer 2 Report
Comments 1: The manuscript describes only the study of metabolites that are secreted into the medium (supernatant was samples of the centrifuged culture). There are also metabolites that are not secreted. According to Wu et al 2022 for example, flavonoids are produced intra and extracellular by Aspergillus. The have different functions. Why were only extracellular flavonoids studied?
Response 1: Thank you for pointing this out. During the experiment, I focused on determining the extracellular flavonoid content. This decision was primarily based on the observation that, during fermentation, the mycelium and secretions of the NM and DG strain groups underwent color changes, with the secretions exhibiting a more pronounced color alteration. Consequently, I prioritized the measurement of extracellular flavonoid compounds. Your suggestion has been extremely valuable. In the subsequent research, I plan to incorporate the intracellular flavonoid content into the study as well to gain a more comprehensive understanding of the relevant phenomena.
Reply 1 reviewer: I haven’t seen in the manuscript that a choice for examining only excreted flavonoids is based on this observation. I would expect that the authors explain this and also emphasize that there might also exist differences in intracellular flavonoid between normal and degenerate lines and that this is subject to a subsequent research.
Comments 2: In line 274-275, the authors mention 1884 metabolites annotated from the metabolome of morel mushrooms. Are these data generated from previous experiments, and if so, by who (ref.)? Add this information into figure 1. Since this manuscript describes metabolites excreted by mycelium, how comparable are mushroom data with the mycelia data?
Response 2: Thank you for pointing this out. These metabolites were annotated using the KEGG database, HMDB database, and LIPIDMaps database. The methodological descriptions are in lines 150 - 151 of the manuscript. The detailed information of the 1884 metabolites is in Table S3.
Reply 2 reviewer: The authors did not answer my question. Are the metabolite categories shown in figure 1d from mushrooms (as stated in the text) and not from mycelia? If so, indicate this in the figure. And if these are indeed from mushrooms, how comparable. useful is this for metabolites from mycelia?
Comments 4: Lines 495-498. The authors hypothesize that a NR-PKS gene is a key gene in the synthesis of flavonoids. Zhang et al 2023. (Discovery of a Unique Flavonoid Biosynthesis Mechanism in Fungi by Genome Mining), mention 2 other key genes typical for fungal biosynthesis of flavonoids: chalsome isomerase and flavone synthase). Are these gene present in the Morchella genome? Also, citing this paper would be useful.
Response 4: Thank you for pointing this out. In the genome of M. importuna, the two above genes were not identified. This NR-PKS shares 36.3% sequence identity with OpS1 from Beauveria bassiana, and both OpS1 and NR-PKS from M. importuna contain domains such as KS, AT, ACP, and TE. Given these similarities and domain compositions, the article should appropriately reference the study by Feng et al. on the OpS1 gene (Fungal biosynthesis of the bibenzoquinone oosporein to evade insect immunity).
Reply 4 reviewer: I think that the reference of Zhang et al 2023 is an important and relevant review and should be included. I also think that the authors should address the absence of the 2 other genes mentioned by these authors and if this is relevant.
Comments 6: Line 479: “ …….. biosynthesis of other secondary metabolites…” Why “other”?
Response 6: Thank you for pointing this out. “Biosynthesis of other secondary metabolites” is the name of a dedicated metabolic pathway in the KEGG database. see the figure below.
Reply 6 reviewer: “..other metabolites….” Is indeed used in the KEGG table but in a text, where this table is not seen, such a remark is odd. Better, “..other than the main metabolites…”
See above
Author Response
Reply 1 reviewer: I haven’t seen in the manuscript that a choice for examining only excreted flavonoids is based on this observation. I would expect that the authors explain this and also emphasize that there might also exist differences in intracellular flavonoid between normal and degenerate lines and that this is subject to a subsequent research.
Response 1: Thank you for pointing this out. Following your suggestions, I have made the revisions to the manuscript, as detailed in lines 276-283 and 552-556.
Reply 2 reviewer: The authors did not answer my question. Are the metabolite categories shown in figure 1d from mushrooms (as stated in the text) and not from mycelia? If so, indicate this in the figure. And if these are indeed from mushrooms, how comparable. useful is this for metabolites from mycelia?
Response 2: Thank you for pointing this out. The previous statement was not very accurate. the 1884 metabolite categories shown in Figure 1d are from mycelia, not from mushrooms. I have already made revisions to the original text, as shown in lines 288-289.
Reply 4 reviewer: I think that the reference of Zhang et al 2023 is an important and relevant review and should be included. I also think that the authors should address the absence of the 2 other genes mentioned by these authors and if this is relevant.
Response 4: Thank you for pointing this out. Following your suggestions, I have revised the manuscript and cited the article by Zhang et al. For details, see lines 513-520 and 659-660.
Reply 6 reviewer: “..other metabolites….” Is indeed used in the KEGG table but in a text, where this table is not seen, such a remark is odd. Better, “..other than the main metabolites…”
Response 6: Thank you for pointing this out. Following your suggestions, I have revised the manuscript. For details, see lines 496-497.